# Synergistic Effects of Nanomedicine Targeting TNFR2 and DNA Demethylation Inhibitor—An Opportunity for Cancer Treatment

**DOI:** 10.3390/cells9010033

**Published:** 2019-12-20

**Authors:** Mohammad A. I. Al-Hatamleh, Engku Nur Syafirah E.A.R., Jennifer C. Boer, Khalid Ferji, Jean-Luc Six, Xin Chen, Eyad Elkord, Magdalena Plebanski, Rohimah Mohamud

**Affiliations:** 1Department of Immunology, School of Medical Sciences, Universiti Sains Malaysia, 16150 Kelantan, Malaysia; si2003@putra.unisza.edu.my; 2Department of Medical Microbiology and Parasitology, School of Medical Sciences, Universiti Sains Malaysia, Kelantan 16150, Malaysia; engkunursyafirah@gmail.com; 3Translational Immunology and Nanotechnology Unit, School of Health and Biomedical Sciences, RMIT University, Bundoora 3083, Australiamagdalena.plebanski@rmit.edu.au (M.P.); 4Université de Lorraine, CNRS, LCPM, F-5400 Nancy, France; khalid.ferji@univ-lorraine.fr (K.F.); jean-luc.six@univ-lorraine.fr (J.-L.S.); 5State Key Laboratory of Quality Research in Chinese Medicine, Institute of Chinese Medical Sciences. University of Macau, Macao 999078, China; 6Cancer Research Center, Qatar Biomedical Research Institute, Hamad Bin Khalifa University, Qatar Foundation, 34110 Doha, Qatar; eelkord@hbku.edu.qa; 7Hospital Universiti Sains Malaysia, Universiti Sains Malaysia, Kelantan 16150, Malaysia

**Keywords:** nanoparticles, immunosuppressive, regulatory T cells, TNF, immunotherapy

## Abstract

Tumor necrosis factor receptor 2 (TNFR2) is expressed on some tumor cells, such as myeloma, Hodgkin lymphoma, colon cancer and ovarian cancer, as well as immunosuppressive cells. There is increasingly evidence that TNFR2 expression in cancer microenvironment has significant implications in cancer progression, metastasis and immune evasion. Although nanomedicine has been extensively studied as a carrier of cancer immunotherapeutic agents, no study to date has investigated TNFR2-targeting nanomedicine in cancer treatment. From an epigenetic perspective, previous studies indicate that DNA demethylation might be responsible for high expressions of TNFR2 in cancer models. This perspective review discusses a novel therapeutic strategy based on nanomedicine that has the capacity to target TNFR2 along with inhibition of DNA demethylation. This approach may maximize the anti-cancer potential of nanomedicine-based immunotherapy and, consequently, markedly improve the outcomes of the management of patients with malignancy.

## 1. Introduction

Tumor necrosis factor (TNF) was first described in the early 1960s as its originally discovered role during tumor regression in rodents [1]. After its initial discovery, many research groups have studied the implications of TNF interactions with its receptors 1 and 2 (TNFR1 and TNFR2) in various inflammatory conditions [2,3]. As inflammation plays a key role in cancer progression and the microenvironment of cancer is controlled by inflammatory cells [4], previous studies have explored TNF-TNFRs interactions in cancer proliferation, metastasis and immune evasion [5]. These studies concluded that TNF works as a tumor-suppressive cytokine through interaction with TNFR1, which is expressed on both normal and tumor cells [6,7,8]. In contrast, TNFR2 expression was only reported on tumor cells and suppressive immune cells, suggesting that TNFR2 directly promotes cancer development [9]. Furthermore, the latest preclinical and clinical studies confirmed the implication of TNF-TNFR2 signaling in cancer proliferation and the suppression of immune response [10,11]. Consequently, blockade of TNF-TNFR2 axis could be a promising option for cancer treatment (Figure 1).

On the other hand, nanotechnology provided several platforms in the field of immunotherapy and made significant advances towards safe and efficient targeting systems with positive clinical outcomes and minimal side effects; this is the so-called field of nanomedicine [12]. Despite the emerging trends of using nanoparticles in immunotherapy, no studies have investigated any type of nanocarriers to target TNFR2 on cancer cells. However, based on some previous novel reports about the effects of nanoparticles on immune homeostasis [13], especially using nanoparticles to promote TNFR2^+^Foxp3^+^ regulatory T cells (Tregs) in inflammatory models [14,15], we have suggested in our recent perspective review the possibility of blocking TNF-TNFR2 axis via nanoparticles [16].

Besides immunotherapy, the use of epigenetic modifications to influence the differentiation of programmed lineage pathways within the immune system has also emerged a few years ago [17]. For example, the inability to differentiate between Tregs and activated T cells neither through FOXP3 mRNA nor protein detection has made it difficult to study these cell subsets in the context of various diseases. To bypass this problem, previous studies have used DNA demethylation within the *FOXP3* gene, to specifically identify Tregs in human peripheral tissue. Since FOXP3 is an essential marker for CD4^+^ Tregs and its expression plays a vital role in controlling the transcriptional program of Tregs, this approach could provide us valuable insight in understanding the molecular features of Tregs in healthy and disease states [18,19]. In other diseases like inflammatory bowel disease, gene expression profile was reported to be related to downregulation of TNFR2, thus promoting CD8^+^ T cell role in these types of inflammatory responses [20].

Furthermore there are large-scale of studies investigating the transcriptomic expressions of several immune checkpoints involved in cancer immune evasion as reviewed by Jamieson and Maker [21] but no studies to date have investigated DNA methylation within the *TNFR2* gene or the transcriptomic expressions of TNFR2 in cancer models. Therefore, to improve the efficiency of cancer immunotherapy, we hypothesized that using DNA demethylation inhibitor along with nanomedicine targeting TNFR2 could be a novel effective approach towards cancer therapy.

## 2. Implication of TNFR2 in Cancer Development

Previous studies have shown that TNF preferentially binds to TNFR2 based on its higher affinity compared to TNFR1 [22,23]. TNFR1 is the primary mediator of TNF-induced apoptosis through its death domain (DD) which activates the nuclear factor kappa B (NF-κB) pathway [24]. TNFR2 also participates in enhancing apoptosis, by activating B cells and plays a crucial role in other pro-inflammatory responses, including proliferation of T cells [2]. Since Chen and his research group discovered TNFR2 for the first time in 2008, many reports have followed up on the potential impacts of TNFR2 expression on cancer cells [25,26,27,28]. As we mentioned before, these studies confirmed that TNFR2 was implicated in proliferation, metastasis and immune evasion of cancer cells by activating immunosuppressive cells.

When TNF-TNFR2 axis is activated, the intracellular domains activate the complexes consisting of TNF receptor-associated factor-2 (TRAF-2), cellular inhibitor of apoptosis protein-1 (cIAP-1) and cIAP2 resulting in the initiation of canonical and non-canonical activation of three main pathways, including NF-kB, activator protein 1 (AP1) and mitogen-activated protein kinases (MAPK) pathways [29,30]. The activity of these pathways both in cancer cells and immunosuppressive cells has led to the conclusion that TNFR2 contributes to cancer progression, expansion as well as stability of Tregs [16]. These pathways activate the phosphoinositide 3-kinases/protein Kinase B pathway (PI3K/Akt) signal transduction pathway that promotes survival and growth [31,32]. Further, NF-kB pathway promotes the transcription of genes responsible to cell proliferation and survival [33]. Activation of PI3K/Akt pathway reduces the differentiation of T helper 17 cells (Th17), which is associated with increased phosphorylation of signal transducer and activator of transcription 5 (STAT5) [33]. The STAT5 got a critical role in the function and activation of Tregs and it is associated with suppression of the anti-tumor activity of Teffs and an increase in proliferation, survival and immune invasion of cancer cells [34]. The suppressor mechanism of STAT5 is based on enhancing the secretion of interleukin-10 (IL-10) and transforming growth factor-β (TGF-β) [35]. NF-kB also leads to increased IL-2 expressions via activating its promoter, which enhances expansion and stability of Tregs that expressing abundant amounts of the IL-2 receptor (IL-2R) associated with improving their suppressor function [36]. The overall description of the implication of TNFR2 activation in cancer cells progression and promoting immunosuppressive cells is summarized in Figure 2 [29,33,37,38].

Tregs express higher levels of TNFR2 than any other immunosuppressive cell, indicating a crucial role of Tregs in cancer microenvironment [39]. Therefore, TNFR2 has emerged in recent years as a potential target of cancer immunotherapy, although the intracellular mechanisms of TNFR2 in cancers are still unclear to some extent [9]. Table 1 shows several studies on the implication of TNFR2 in cancer development and suppressive immune responses. It is worth mentioning that, although several anti-cancer immunotherapeutics have been approved by the Food and Drug Administration (FDA) [12], to date there is no drug approved by the FDA as an TNFR2-antagonist.

## 3. Nanomedicine Applications for Cancer Immunotherapy

The introduction of advanced nanotechnology in the medical field aimed to better prevention, diagnostics and therapy of diseases, is called nanomedicine [41]. Incorporating therapeutic drugs with nano-objects (NOs) showed promising results in avoiding systemic side effects of drugs by improving its precision in targeting at the cellular level [42,43]. Most of the NOs are objects with a spherical shape but all the NOs are with nanometric size (most often below 200 nm) [44]. Thanks to their small size, relatively close to that of proteins, these NOs easily cross the different barriers immunity. Moreover, according to their chemical composition, these NOs are able to encapsulate and protect hydrophilic or/and hydrophobic drugs against any enzymatic, chemical or physical degradation, all along their transport to the target cells [44]. The NOs currently used in the biomedical field are complex systems that can be made of various components according to the desired application. Researchers utilized several types of drug-loaded NOs (nanocarriers) to enhance cancer therapy, including organic, inorganic and hybrid nanoparticles (NPs) (Table 2) [45]. Nanomedicine provided many effective approaches in cancer immunotherapy, while nano-based drug delivery has received considerable interest as well (Figure 3).

Owing to their properties as effective antigen-presenting cells (APCs), dendritic cells (DCs) have been utilized in cell-mediated therapeutic vaccination via several strategies to present antigens against cancer cells [57]. Therefore, it was hypothesized that targeting DCs with antigens and adjuvants loaded with NPs might be an effective strategy to adopt the use of DCs in cancer immunotherapy [58,59]. So far, one study has successfully used a poly-lactic-co-glycolic acid (PLGA)-based NPs in targeting DCs via biotinylated antibodies and it was enhanced T cell immunity [60]. In this study, targeting more DCs subsets was significantly correlated with more T cell activation [60].

Utilizing NPs to change tumor microenvironment can produce an immunosuppressive environment based on the combination of NPs with several immunotherapy applications, including chemotherapy, cytokines and many biomolecules towards releasing soluble cytokine mediators and attracting immunosuppressive cells [12]. A study on advanced melanoma model showed that combining of polyethylene glycol (PEG) conjugated curcumin and vaccine, dramatically decreased the number of immunosuppressive cells and reduced the levels of IL-6 and chemokine ligand 2, while showing an increase in levels of proinflammatory cytokines and elevation in the CD8^+^ T-cells [61]. In gene delivery, several NPs have been effectively utilized to deliver small interfering RNA (siRNA) mainly to TGF-β, cytotoxic T-lymphocyte–associated antigen 4 (CTLA-4), programmed death-ligand 1 (PD-L1) and PD-L2. These approaches showed a significant inhibition in cancer development due to increase of CD8^+^ T cells and decreased of immunosuppressive cells [62,63,64].

Antibody-based treatments are a promising approaches in cancer immunotherapy and emerged due to their high degree of specificity to target precision sites implicated in cancer development [65]. However, there is a need for novel delivery strategies to reach the highest levels of specificity in antibody targeting, while trying to avoid off-target side effects as much as possible. Antibody delivery has been reported to be the most common nanomedicine application in terms of immunotherapy [66]. NPs have been utilized in co-delivery of several antibodies to blockade CTLA-4, PD-1 and PDL-1 in different cancer models [67,68]. It also successfully improved the immunotherapeutic antibodies inhibit cancer’s immunosuppressive checkpoint pathways, thus inhibiting Tregs, activation of Teffs and fight cancer cells [69,70]. Although the promising findings were reported by studies of antibody-based drugs which have been incorporated with NPs to target significant receptors on cancer cells and immune cells in cancer microenvironment, no studies to date have combined TNFR2 antibodies with NPs towards an effective immunotherapy approach. Therefore, we hypothesized that the utilization of NPs specific ligands to delivery TNFR2 antibodies would alter immunological status of cancer microenvironment; results in fight cancer cells and immunosuppressive cells specifically (Figure 4).

## 4. DNA Demethylation and Immune Evasion

The 5-methylcytosines within CG dinucleotides (CpG islands) are establishing to keep the balance of normal development and this process called DNA methylation [71]. As cancer progression results in many modifications in the normal mechanism of gene expression, demethylation of DNA is emerged to be one of the critical mechanisms that leads to modify gene expression and implicated in cancer progression, metastasis and immune evasion [72]. The overall description of the mechanisms of DNA methylation and demethylation are summarized in Figure 5. Although the critical role of TNFR2 in immune evasion of tumor cells, no study to date has been investigated on DNA methylation/demethylation in the *TNFR2* gene in cancer subjects. Whereas there were significant findings have been reported recently in studies investigated on methylation/demethylation in several immune checkpoint genes including *PD-1*, *PD-L1*, *CTLA-4*, *TIM-3*, *LAG-3* and *TIGIT* [73,74]. Demethylation and many mechanisms are part of the epigenetics field, which is considered as a gate for promising trends in cancer research in the current era.

It has been reported that epigenetic regulation is one of the key mechanisms which facilitate cancer progression and immunosuppression, including the regulation of immune checkpoints expression in the tumor microenvironment [76]. For example, three important epigenetic modifications have been reported in colorectal cancer; DNA methylation, post-translational modifications in chromatin-protein interactions and expression of non-coding RNAs [77,78]. Silencing of tumor suppressor genes can occur as a result of hypermethylation of the CpG islands, which highly enriched within the promoter regions of tumor suppressor genes [79]. Alternatively, promoter demethylation and distribution of repressive histones can work together to induce the upregulation of many genes in cancers [80]. Besides, it has been suggested that the enrichment of repressive histones, histone 3 lysine 9 trimethylation (H3K9me3) and H3K27me3 in the promoter regions along with CpG hypermethylation are the common epigenetic modifications in breast cancer [74]. Therefore, we hypothesized that *TNFR2* gene might be highly expressed and demethylated in cancer models. Thus using demethylation inhibitors as a potential approach will result in decrease of the relative expression of *TNFR2* gene; consequently, downregulation of TNFR2 on cancer cells and immune suppressive cells. Furthermore, using the above hypotheses in the meanwhile with TNFR2 antibodies delivered via NPs is novel hypotheses for an effective cancer immunotherapy (Figure 6).

Although the co-administration of TNFR2 antibodies conjugated with NPs used synergistically with DNA demethylation, would be a promising therapeutic approach, the clinical usage of these agents is likely to require further investigation to establish dosing, injection procedure and administration patterns. Although this is considered a relatively novel approach, we do not expect any significant side effects for TNFR2 targeting via this approach, as the receptor is preferentially expressed on cancer cells and immunosuppressive cells. Furthermore, previous studies aimed to modulate TNFRs by targeting TNF itself, which might have caused total inhibition of TNF and thus the onset of serious side effects including peripheral and central nervous system demyelinating disorders [81]. In addition, other well-known side effects could emerge, such as injection site reactions, hemocytopenia, congestive heart failure and T-cell lymphomas [82]. Therefore, we believe our perspective provides a relatively safer and more targeted approach towards an effective cancer treatment.

## 5. Conclusions and Future Directions

It is known that the TNF works as an anti-tumor cytokine, while its role may be negatively converted to be a tumor-promoting substance upon its conjugation with TNFR2, which is only expressed on some tumor cells and immunosuppressive cells. In both cell types, TNF-TNFR2 axis was implicated in increasing growth and development via three main pathways, including NF-kB, AP1 and MAPK pathways. Based on these pathways, TNF-TNFR2 axis was responsible for promoting carcinogenesis and cancer immune evasion. Consequently, there were several studies showing blockade of TNF-TNFR2 axis as a promising cancer treatment. Towards an efficient and precision immunotherapy, nanomedicine provided many effective approaches, especially for antibodies-based drugs delivery, not only to reach the targeted sites but also to restore the immune response to suppress the cancer cells. To date, no study has employed any nanomedicine in blockade TNFR2.

Moreover, based on the understanding that significant epigenetic modifications occur in cancer, recent epigenetic reports provided new insight to improve responses to immunotherapy targeted immune checkpoints via specific epigenetic drugs. One of the most common and significant modifications is DNA demethylation. However, no study to date has been investigated on DNA methylation/demethylation of *TNFR2* gene. Therefore, future studies should investigate how to incorporate TNFR2 antibodies in nanoparticles with specific characterization to create a more efficient blockade for TNFR2. Studies should also focus on understanding the status of DNA methylation/demethylation pattern in the promoter region of *TNFR2* gene, that will be helpful before using DNA demethylation inhibitors. In the end, providing a combination of nanomedicine-based immunotherapies (TNFR2 antibodies) at the same time with epigenetic drugs (DNA demethylation inhibitors) will be a novel strategy to improve the effectiveness of cancer immunotherapies.

## Figures and Tables

**Figure 1 cells-09-00033-f001:**
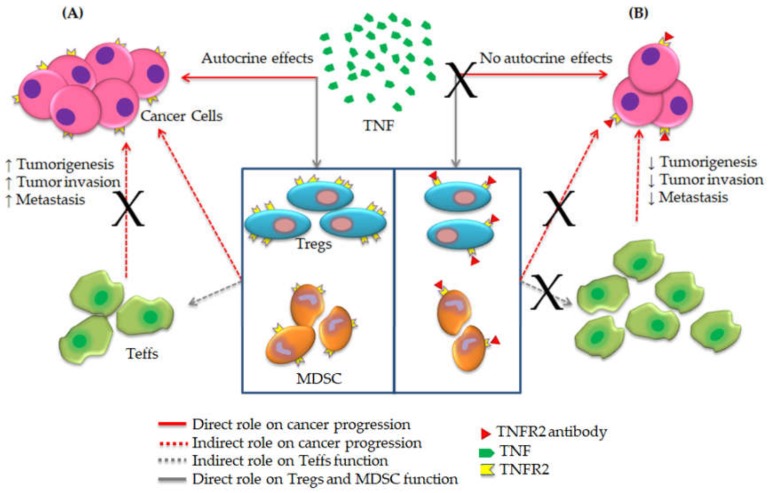
The blockade mechanism of TNF-TNFR2 axis towards effective cancer immunotherapy. (**A**) TNFR2 is expressed on cancer cells and suppressive immune cells, including regulatory T cells (Tregs) and myeloid-derived suppressor cells (MDSCs). Binding of TNF to TNFR2 enhances cancer proliferation and immune evasion by expansion of Tregs and MDSCs along with the suppression of the immune response of effector T cells (Teffs). (**B**) Antibody blockade of TNFR2 inverts this mechanism and promotes effector functions against cancer (Amended from Al-Hatamleh et al., 2019 [16]).

**Figure 2 cells-09-00033-f002:**
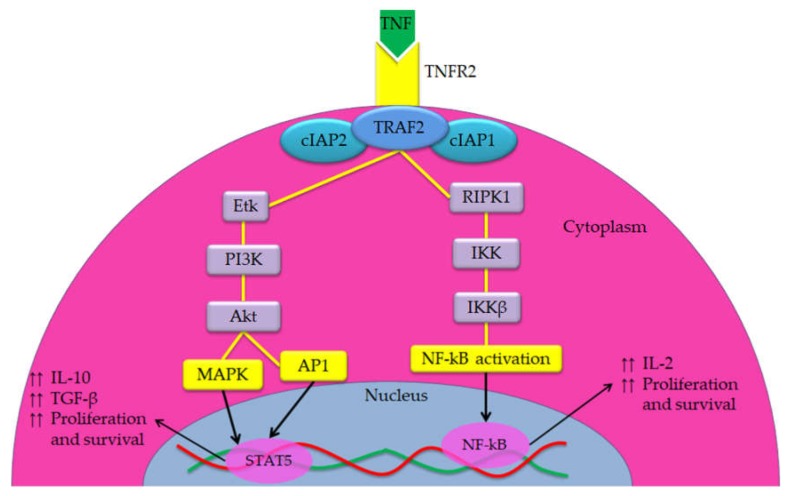
Overview of the TNF-TNFR2 signaling pathway. In both cancer cell and immunosuppressive cells, TNFR2 is activated by both soluble TNF (sTNF) and membrane-bound TNF (mTNF) but it is fully activated by mTNF. TNFR2 does not interact with an intracellular DD, while it interacts with complex I that consists of TRAF2 with cIAP1 and cIAP2 and induction of homeostatic signals. The signals travel from complex I either via receptor-interacting serine/threonine-protein kinase 1 (RIPK1) or Etk (a member of the Btk tyrosine kinase family). RIPK1 trigger NF-κB via the IkB kinase (IKK) complex, which results in increasing the transcription of several genes associated positively with cell survival and proliferation. However, the Etk, through the PI3K/Akt pathway, is able to activate both AP1 and MAPK signaling pathways, which activate the promoter of proliferation, survival and other transcription factors. Further, it is associated with enhancing the phosphorylation of STAT5 that play a crucial role in immunosuppression.

**Figure 3 cells-09-00033-f003:**
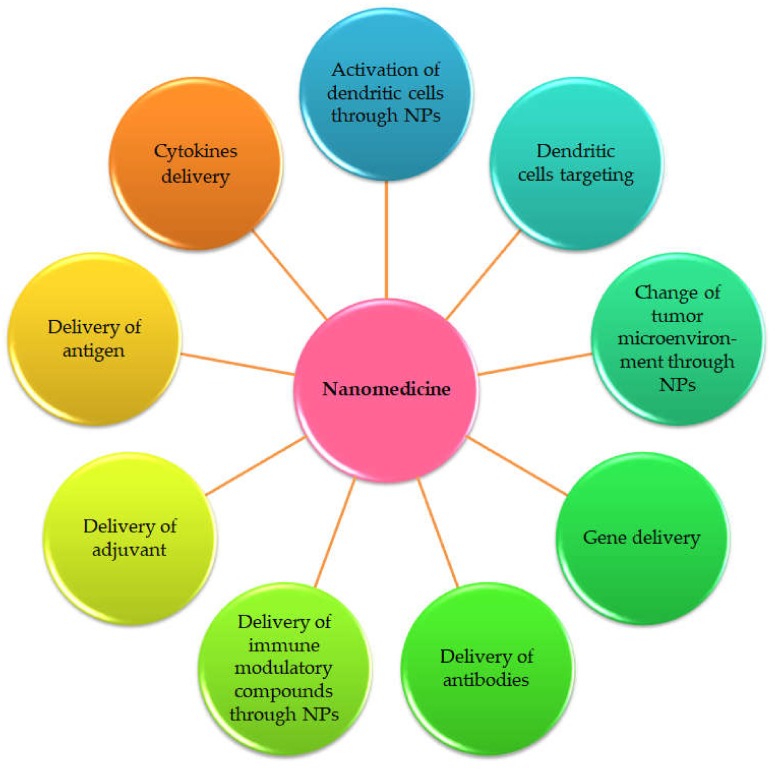
Nanomedicine strategies to improve cancer immunotherapy as suggested by Qiu et al., 2017 [12].

**Figure 4 cells-09-00033-f004:**
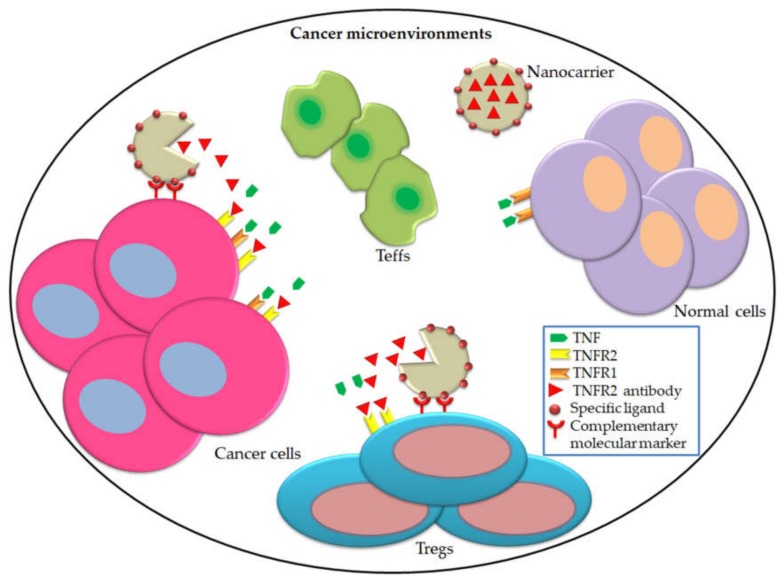
Nanoparticles-targeting TNFR2 exhibit their effects on cancer cells and Tregs. We hypothesized that based on the ability of nanoparticles with specific ligands to targeting cancer cells, it is possible to incorporate TNFR2 antibodies with this kind of nanoparticles. Consequently, TNFR2 antibodies will be released at the cancer microenvironment, which is in turn, will blockade TNFR2 on cancer cells and Tregs. This modulation will result in suppressing Tregs and activate Teffs, thus less tumorigenesis, tumor invasion and metastasis.

**Figure 5 cells-09-00033-f005:**
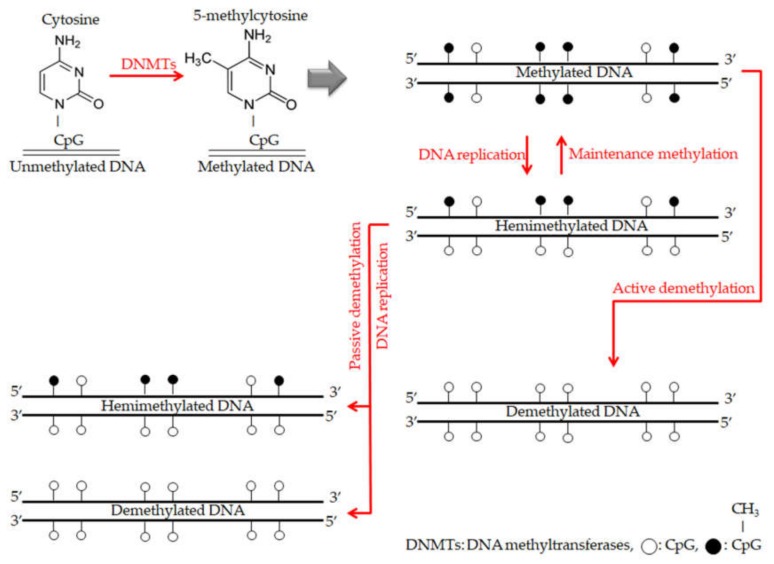
The overall description for mechanisms of DNA methylation and demethylation. During DNA methylation, DNA methyltransferases (DNMTs) enzymes are responsible for adding methyl groups to specific CpG islands and establishing DNA methylation patterns. With each DNA replication round, there is a new strand of unmethylated DNA. Thus the new double strand DNA becomes hemimethylated. Furthermore, maintenance methylation process by DNMT1 enzyme leads to copy that DNA methylation pattern from the old strand onto the new strand. On the other hand, during DNA demethylation, maintenance methylation process finishing with remarkably failing, which leads to stopping DNA methylation patterns (passive demethylation). While, active demethylation achieving by specific enzymes have not been explicitly explored and results in modifying methylated cytosines to only cytosines without methyl groups and independently of DNA replication [75].

**Figure 6 cells-09-00033-f006:**
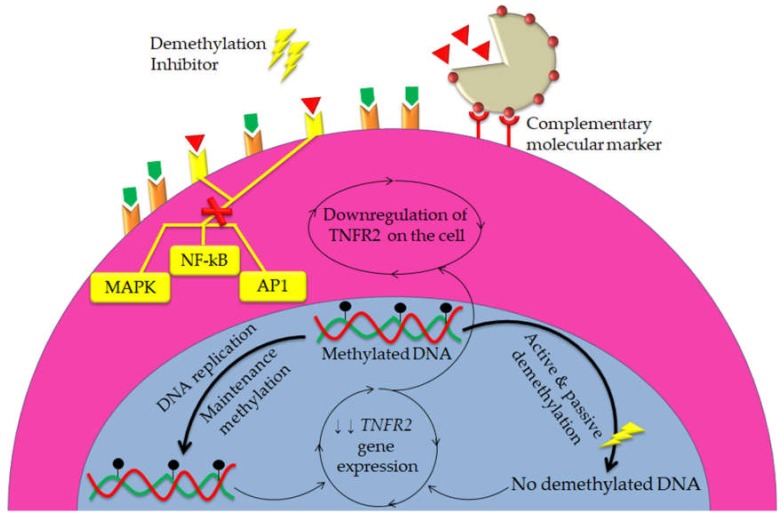
NPs are hypothesized to be utilized synergistically with DNA demethylation inhibitors to stronger blockade of TNFR2 signals in both cancer cells and immunosuppressive cells. Owning to their ability to target specific cellular sites, NPs with specific ligands can deliver TNFR2 antibodies to cancer cells and immunosuppressive cells only, which in turn stop TNFR2 intracellular signal pathways. Meanwhile, using DNA demethylation inhibitors might lead to epigenetic alteration results in decrease the expression of *TNFR2* gene, thus downregulation of TNFR2. Consequently, combination of these two strategies might extend the anti-tumor effects of TNFR2 antibodies with more precision in immunotherapy.

**Table 1 cells-09-00033-t001:** List of studies that directly investigated the significant role of TNFR2 in cancer development and immune evasion.

Study	Samples Used and Diagnosis Assay	Significant Findings
Yan et al., 2015 [25]	- Peripheral blood samples were obtained from cancer patients and healthy controls. - Phenotypic characteristics of TNFR2^+^ Tregs were determined by flow cytometry. - The functional of TNFR2^+^ Tregs was determined by in vitro Treg suppression assay.	- Expression of TNFR2 appeared to correlate with FoxP3 expression. - Increase TNFR2 expression levels on Tregs were positively associated with more advanced clinical stage of cancer, immune invasion and progressive metastasis.
Williams et al., 2016 [40]	- Leukapheresis samples were obtained from healthy donors. - The samples were subjected membrane protein expression arrays, ELISAs and few other assays that are related to the functional of TNFR2 target.	- TNFR2 expressed by Tregs more than Teffs in both cancer models. - TNFR2-specific agonists increased IFN-γ secretion by CD8^+^ T cell and suppressed tumor growth in the mice.
Zhang et al., 2017 [27]	- Peripheral blood and tissue biopsies were taken from patients. - Levels of circulating Tregs were determined by flow cytometry. - Soluble TNFR1 and TNFR2 in plasma were determined by ELISA method. - mRNA expression levels of TNF, TNFR2 and FoxP3 were determined by real-time PCR.	- TNFR2^+^ Tregs significantly increased in cancer patients and inversely correlated with the clinical cancer stages. Their results showed that patients with stage 1 cervical cancer displayed higher percentage of TNFR2^+^ Tregs compared with stage 2. This is probably due to many TNFR2^+^ Tregs have undergone trafficking to the in situ tumor microenvironment from the peripheral circulation as the carcinoma progressed, leading to a decrease in the circulating subsets. - The mRNA expression of Foxp3 and TNFR2 increased in cancer patients.
Torrey et al., 2017 [11]	- Peripheral blood samples were taken from donors. - Ascites samples from ovarian cancer were obtained from newly diagnosed patients. - These samples were used in Tregs functional assays.	- TNFR2 antibodies inhibited Tregs proliferation and enabled Teffs expansion. - The efficiency of TNFR2 antagonists in inhibiting Tregs was stronger with cells isolated from cancer cells culture compared to others from a healthy donor. - The TNFR2 antibodies killed cancer cells.
Nie et al., 2018 [28]	- CT26 tumor cells were subcutaneously injected into the recipient mice and the biopsies were taken after recommended times. - The proportion of Tregs and TNFR2^+^ was determined by flow cytometry.	- Using a combination of TNFR2-blocking antibody with CD25-targeted antibody suppressed cancer growth.

**Table 2 cells-09-00033-t002:** Drug delivery systems for anti-tumor drugs.

Nanocarrier Type [Size Range]	Significant Properties	Selected Studies
**Organic nanocarriers**		
Solid lipid nanoparticles (SLNs) [50–200 nm]	Economical large-scale production, high drug payload, better stability and easy to handle, improved bioavailability of poorly water-soluble drugs, as well as lack of biotoxicity.	In 2017, Wang et al. have designed resveratrol-loaded SLNs (Res-SLNs) to treat breast cancer cells. They showed that Res-SLNs significantly exhibited the inhibitory effects on cancer cells proliferation, invasion and migration, compared to controls [46].
Liposomes and polymersomes [30 nm–110 nm]	Enhanced delivery of drugs, preventing early degradation of the encapsulated drug, cost-effective formulations of expensive drugs and efficient treatment, improved performance features of the product, protection of the active drug from environmental factors, as well as reduced systemic toxicity. Weak mechanical properties of liposomes may be enhanced by using polymeric liposomes called polymersomes [47]. Possibility to encapsulate hydrophobic drugs in the lipidic bilayer, as well as hydrophilic drugs in the hydrophilic core [48].	Doxorubicin (DOX)-loaded specific monoclonal antibodies conjugated to liposomes were used to treat lung tumor in mice. The results showed significant suppression of tumor growth, metastatic spread and increased the survival rate of the tumor-bearing mice compared to controls [49].
Dendrimers [1.5–14.5 nm]	The most utilized nanocarrier owing to their incomparable characteristics, including the increased number of branching, distinctive molecular weight, monodispersed macromolecules, multivalency and spherical shapes.	In treating of lung metastasis mouse model, as indicated by increased survival rates and decreased tumor burden, DOX conjugated to carboxyl-terminated poly(amidoamine) dendrimers (PAMAM) was more effective than DOX delivered intravenously [50].
Polymeric nanoparticles (PNPs) [10–200 nm]	Two types: - Nanospheres PNPs (matrix-type); disperse the drug in the polymer matrix. - Nanocapsules PNPs (reservoir-type); dissolve the drug in aqueous or oily liquid covered by a solid polymeric membrane. PNPs are highly versatile based on a wide range of polymers from synthetic and natural sources. The modification of the physicochemical properties of the polymers used to produce PNPs can accurately control the degradation of PNPs and drug release.	Nanoscale coordination polymer-1 (NCP-1) has been used for simultaneous delivery of Oxaliplatin and Gemcitabine monophosphate. NCP-1 particles effectively avoided uptake by the mononuclear phagocyte system (MPS), which is resulted in potently delivery for both drugs and thus a strong synergistic therapeutic effect was observed against pancreatic cancer cells by inhibiting tumor growth [51].
Polymeric micelles (PMs) [10–100 nm]	PMs allow hydrophobic drugs to be entrapped into their cores which enhance their water solubility. The hydrophilic shell of PMs promotes their stability and their circulation times in blood by preventing the recognition and subsequent uptake of it by the reticule endothelial system.	Pluronic micelles combined with polyplexes spontaneously were used as amphiphilic-based gene delivery system with two breast cancer cell lines. This system has been formed by electrostatic interaction between cationic polyethyleneimine and anionic siRNA against AKT2. After treatment, a significant reduction was observed on cell invasion capacity, as well as a significant inhibition of mammosphere formation [52].
Virus-based nanoparticles (VNPs) [up to 100 nm]	VNPs emerged based on their easy surface functionalization, availability in a variety of sizes and shapes, in addition to their biocompatibility and morphological uniformity.	To overcome immunological tolerance against human epidermal growth factor receptor 2 (HER2) in breast cancer mouse model, HER2 epitopes were integrated on the plant-produced vaccination platform potato virus X (PVX). The results showed that this carrier stimulated the production of HER2-specific antibodies in the injected mice [53].
**Inorganic nanocarriers**		
Carbon nanotubes (CNTs) [0.4–100 nm]	Able to cross the cell membrane via endocytosis and subsequently enter into the cell; based on their needle-like shape. With its physicochemical characteristics, CNTs able to carry high drug amounts, structural flexibility and intrinsic stability and appropriate surface functionalization.	In 2010, Sun et al. have conjugated MCF7 breast cancer cells−derived tumor lysate (covalent) to Carboxylated MWNT (CNTs) *in vitro*. They reported increased antigen uptake by dendritic cells (DCs) and improved the induction of tumor−specific T cell response by DCs, thus enhance the uptake of tumor antigens [54].
Mesoporous silica nanoparticles (MSNs) [2–50 nm]	Owing to their honeycomb-like shape with hundreds of pores, MSNs able to load large drugs amounts. Based on their ease of surface functionalization for targeted and controlled drug delivery, MSNs reduce the toxicity of drugs and promote therapeutic efficacy.	Guo et al. have used MSNs with nuclear targeting in cancer therapy for multidrug resistance (MDR) breast cancer cells. They used a size changeable MSNs able to alter to smaller micelles under specific conditions. This study reported this type of MSNs as a highly effective delivery system for anticancer drugs to the nucleus of MDR cancer cells, directly [55].
**Hybrid nanocarriers**	A combination of organic and inorganic NPs. This combination successfully employed specific functionalities of both NPs to enhance the selectivity and efficiency of drugs along with high payload sustained and intracellular delivery.	A multifunctional hybrid nanocarrier was developed by merging the properties of pH-sensitive nanogels and multiwall carbon nanotube, to deliver the DOX. This study showed a significant effect of DOX supernatant with this hybrid nanocarrier on the U-87 glioblastoma cancer cells proliferation suppression [56].

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
