# Peer review of "Synergistic Effects of Nanomedicine Targeting TNFR2 and DNA Demethylation Inhibitor—An Opportunity for Cancer Treatment"

_cells, 2019, doi:10.3390/cells9010033_

Round 1

Reviewer 1 Report

In this perspective, the authors discussed a possible therapeutic approach by combining a TNFR-2 targeting nanomedicine along with DNA Demethylation inhibitor. This seems to be a very good and intriguing approach and might add new opportunities in the already existing nanomedicine based anti-cancer immunotherapies. The authors discussed and reviewed some papers based on the implication of TNFR-2, nanomedicine, and DNA Demethylation in cancer immunotherapy separately; however, a concerted discussion is missing here. The authors discussed possible positive impacts of this targeting approach in the end, but a discussion is also necessary on possible side effects of this approach based on the already available published articles or clinical trials. I believe this little tweaking will make this nice perspective even more interesting. 

Thanks.

Author Response

Thank you for your notes, we have responded as below:

"Although the co-administration of TNFR2 antibodies conjugated with NPs used synergistically with DNA demethylation, would be a promising therapeutic approach, the clinical usage of these agents is likely to require further investigation to establish dosing, injection procedure and administration patterns. Although this is considered a relatively novel approach, we do not expect any significant side effects for TNFR2 targeting via this approach, as the receptor is preferentially expressed on cancer cells and immunosuppressive cells. Furthermore, previous studies aimed to modulate TNFRs by targeting TNF itself, which might have caused total inhibition of TNF, and thus the onset of serious side effects including peripheral and central nervous system demyelinating disorders [81]. In addition, other well-known side effects could emerge, such as: injection site reactions, haemocytopenia, congestive heart failure, and T-cell lymphomas [82]. Therefore, we believe our perspective provides a relatively safer and more targeted approach towards an effective cancer treatment."

Mohan, N.; Edwards, E.T.; Cupps, T.R.; Oliverio, P.J.; Sandberg, G.; Crayton, H.; Richert, J.R.; Siegel, J.N. Demyelination occurring during anti-tumor necrosis factor alpha therapy for inflammatory arthritides. Arthritis Rheum 2001, 44, 2862-2869, doi:10.1002/1529-0131(200112)44:12<2862::aid-art474>3.0.co;2-w. Kemanetzoglou, E.; Andreadou, E. CNS Demyelination with TNF-alpha Blockers. Curr Neurol Neurosci Rep 2017, 17, 36, doi:10.1007/s11910-017-0742-1.

Reviewer 2 Report

In this minireview, Mohammad A. I. Al-Hatamleh et al, discusses a novel therapeutic strategy based on nanomedicine that has the capacity to target TNFR2 along with inhibition of DNA demethylation.

They first introduced the TNF/TNFRs in the context of inflammation, microenvironment and cancer development and progression, discussing on the effects mediated by this specific signalling in selected immunosuppressive cells.

The authors also briefly described nanomedicine applications for cancer immunotherapy and several features of epigenetic control of immune evasion pathways.

They conclude that “providing a combination of nanomedicine-based immunotherapies (TNFR2 antibodies) at the same time with epigenetic drugs (DNA demethylation inhibitors) will be a novel strategy to improve the effectiveness of cancer immunotherapies”.

Major points:

This review is too generic and didactic; it doesn’t deepen and discuss the literature concerning these issues and only provides generic short descriptive concepts of limited usefulness to the readers.

Author Response

Thank you very much,
We will improve our manuscript accordingly. 

Reviewer 3 Report

This is an interesting and robust review with engaging content and diagrams. I do not feel that I could suggest anything which may improve the manuscript and for that reason I am recommending acceptance in its current form.

Author Response

Thank you very much

Reviewer 4 Report

I find the paper very well written, with very nice illustrations and reporting on a very important topic. In my opinion, it can be accepted as it is.

Author Response

Thank you very much